# Impact of the Strong Downwelling (Upwelling) on Small Pelagic Fish Production during the 2016 (2019) Negative (Positive) Indian Ocean Dipole Events in the Eastern Indian Ocean off Java

**Jonson Lumban-Gaol** [1,*]**, Eko Siswanto** [2]**, Kedarnath Mahapatra** [3]**, Nyoman Metta Nyanakumara Natih** [1]**, I Wayan Nurjaya** [1]**, Mochamad Tri Hartanto** [1]**, Erwin Maulana** [1]**, Luky Adrianto** [4]**, Herlambang Aulia Rachman** [5]**, Takahiro Osawa** [6]**, Berri Miraz Kholipah Rahman** [7] **and Arik Permana** [8]

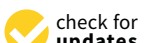

1   Department of Marine Science and Technology, Faculty of Fisheries and Marine Science, IPB University, Bogor 16680, Indonesia; natih@apps.ipb.ac.id (N.M.N.N.); i.wayan.nurjaya@apps.ipb.ac.id (I.W.N.); mochamadha@apps.ipb.ac.id (M.T.H.); erwin.itk@gmail.com (E.M.)
2   Earth Surface System Research Center, Research Institute for Global Change, Japan Agency for Marine-Earth Science and Technology, 3173-25, Showa-machi, Kanazawa-ku, Yokohama, Kanagawa 236-0001, Japan; ekosiswanto@jamstec.go.jp
3   School of Marine Science and Technology, Tokai University, 3-20-1 Orido, Shimizu 424-8610, Japan; kedar@scc.u-tokai.ac.jp
4   Department of Aquatic Resources Management, Faculty of Fisheries and Marine Science, IPB University, Bogor 16680, Indonesia; lukyadrianto@ipb.ac.id
5   Marine Technology, Faculty of Fisheries and Marine Science, IPB University, Bogor 16680, Indonesia; herlambangauliarachman@gmail.com
6   Center for Remote Sensing and Ocean Science (CReSOS), Udayana University, Post Graduate Building, Denpasar 80232, Indonesia; osawa320@gmail.com
7   Marine Fisheries Technology, Faculty of Fisheries and Marine Science, IPB University, Bogor 16680, Indonesia; berri_miraz13@apps.ipb.ac.id
8   Faculty of Fisheries and Marine Science, IPB University, Bogor 16680, Indonesia; arik.ikan@gmail.com
*   Correspondence: jonsonlumban@apps.ipb.ac.id

**Abstract:** Although researchers have investigated the impact of Indian Ocean Dipole (IOD) phases on human lives, only a few have examined such impacts on fisheries. In this study, we analyzed the influence of negative (positive) IOD phases on chlorophyll a (Chl-a) concentrations as an indicator of phytoplankton biomass and small pelagic fish production in the eastern Indian Ocean (EIO) off Java. We also conducted field surveys in the EIO off Palabuhanratu Bay at the peak (October) and the end (December) of the 2019 positive IOD phase. Our findings show that the Chl-a concentration had a strong and robust association with the 2016 (2019) negative (positive) IOD phases. The negative (positive) anomalous Chl-a concentration in the EIO off Java associated with the negative (positive) IOD phase induced strong downwelling (upwelling), leading to the preponderant decrease (increase) in small pelagic fish production in the EIO off Java.

**Keywords:** chlorophyll-a; climate change; IOD; Palabuhanratu Bay; pelagic fishery; sea surface temperature

## 1. Introduction

The Indian Ocean Dipole (IOD) is well-known as a dominant mode of interannual climate variability that develops from air–sea interactions in the Indian Ocean. With anomalously low sea surface temperatures (SST) associated with strong upwelling in the eastern Indian Ocean (EIO) off Java–Sumatra and high SST in the western Indian Ocean, it is known as the IOD positive phases (pIOD). Featuring opposite anomalies over a similar region are the IOD negative phases (nIOD) [1–3]. The pIOD events have become stronger and more frequent, particularly since the 1960s [4,5]. The publications by Abram et al. [4]

and Cai et al. [5] indicate that the change in IOD occurrences can be attributed to climate change and to greenhouse warming [6].

Researchers have widely investigated the impact of IOD phases on human lives. They have found it to be considerably large, particularly with respect to socioeconomics. For example, the IOD plays an important role in monsoon rainfall. The rainfall over East Africa (Indonesia) increases (decreases) during a pIOD phase [7]. High rainfall during the pIOD serves as a driving force in the resurgence of malaria in the East African highlands [8,9]. Furthermore, there have been reports on the impact of IOD on southeast Australian bushfires [10] and Zimbabwean droughts [11].

The impact of the IOD on the physical structure of the Indian Ocean is well known. The stronger-than-normal seasonal southeasterly winds along the Java–Sumatra coasts during the pIOD phase have intensified coastal upwelling [12]. During the pIOD phase, the intense upwelling event occurs concurrently with a chlorophyll a (Chl-a) bloom off the South Java and Sumatra coasts [13–15]. Meanwhile, during the 2016 nIOD, the downwelling event occurred concurrently with a Chl-a decrease along the coasts of Sumatra and Java [16].

Although many researchers have tried to understand IOD impacts, only a few have examined such impacts on small pelagic fisheries. Generally, during the pIOD, fish production increased sharply in the EIO owing to increased upwelling [17–19], but nIOD phase impacts on fish production are not well known in the EIO off Java.

According to mass media information, small pelagic fish landing in the fishing ports tended to decrease during negative IOD phases from 2016 to 2017. Conversely, it tended to increase during the 2019 pIOD. The high fluctuation in the fish landing during the IOD phases has influenced socio-economic activities. Usually, if the quantum of fish landing decreases, the price of fish increases, and as a result, the raw materials for the fishing industry reduce. However, the quantity of these raw materials increases during an increase in fish landing.

The Dipole Mode Index (DMI) shows that 2016 and 2019 could be two of the most extreme nIOD and pIOD years ever, respectively [20,21], as shown in Figure 1. The negative and positive impacts of this phenomenon need to be investigated to enhance understanding with respect to small pelagic fishery production and to predict of future productions.

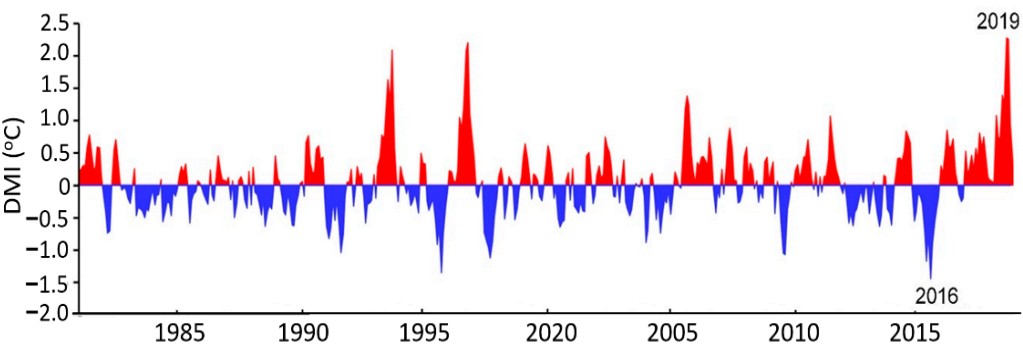

**Figure 1.** Dipole Mode Index showing the occurrence of the negative phases (blue) and positive phases (red) of Indian Ocean Dipoles during 1980–2019. The negative and positive extreme Indian Ocean Dipoles occurred in 2016 and 2019, respectively, marked with the years of pronounced negative and positive anomalies.

## 2. Materials and Methods

We conducted field surveys twice a year, at the peak of the pIOD (5 October 2019) and the end of the pIOD (18 December 2019) phases, by deploying and water-sampling Conductivity Temperature Depth (CTD) sensors at five stations (St-1: 7.00° S 106.45° E, St-2: 7.03° S 106.45° E, St-3: 7.05° S 106.42° E, St-4: 7.07° S 106.38° E, St-5: 7.12° S 106.31° E) in the EIO off Palabuhanratu Bay (Figure 2). This bay is the largest in the EIO off West Java, and is geographically located at 6.90° to 7.10° S and 106.30° to 106.50° E, with a coastline of approximately 105 km. The bathymetry of Palabuhanratu Bay is steep, with a depth of

between 3 and 4 m (coastal estuary waters) to more than 200 m in the central part of the bay waters where it is a continental slope. Such a topographic profile results in the longshore current in several locations of the bay. At one point, there was a depth of 200 m at about 2 km off the coast. At another point, a rapid increase in depth supposedly formed a large puddle with a width of about 300 m. The depths of more than 500 m are in locations 6.5 km off the coast.

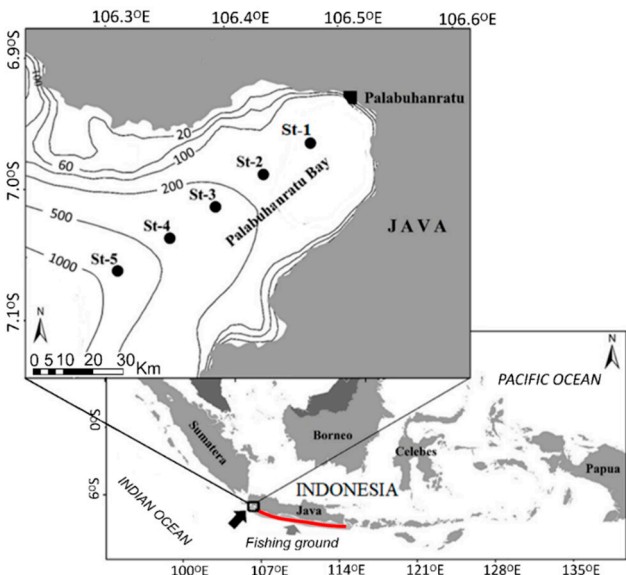

**Figure 2.** The location map of the southern coast of West Java showing Palabuhanratu Bay (inset) and the five sampling stations (St-1–5) in the bay, and the line (red) along the fishing ground off the southern coast of Java used for plotting the Hovmöller diagram of satellite-derived sea surface temperature and chlorophyll-a concentration.

We analyzed the in-situ Chl-a concentrations from water samples at each station using the American Public Health Association (APHA) standard method (APHA, 23rd Edition, 10200-H, 2017) both at the peak and end of the pIOD.

To elucidate the difference in the Chl-a distributions between the nIOD (2016) and pIOD (2019) phases, we used the daily AQUA-Moderate Resolution Imaging Spectro-radiometer (MODIS) Chl-a concentration data at 1 km resolution from July to October (southeast monsoon period), and the data source was the National Aeronautics and Space Administration (NASA) Earth Data Open Access for Sciences [22]. The monthly means of AQUA-MODIS Chl-a concentration and SST data at 4 km resolution from 2003–2019 period used in this study were from the NASA Goddard Space Flight Center, Ocean Ecology Laboratory, Ocean Biology Processing Group [23].

We plotted the Hovmöller diagram to demonstrate Chl-a and SST variability during the pIOD and nIOD phases during the 2003–2019 period, and the power spectral density (PSD) analysis was carried out to determine the energy dominant temporal variability of Chl-a concentrations [24]. The IOD mode index dataset used in this study was from the NOAA Physical Sciences Laboratory, Boulder, Colorado, U.S.A. [25]. For sea surface elevation, we used the Hybrid Coordinate Ocean Model (HYCOM) 1/12-degree GOFS 3.1 analysis data provided by the Asia-Pacific Data Research Center [26].

We conducted the fish-landing observations at the Palabuhanratu fishing port in October and December 2019. We collected the monthly small pelagic fish catch data to compare fish catches between the 2016 nIOD (2016) and nIOD (2019) phases. To compare the catches between the nIOD and pIOD phases, we performed a paired Student's *t*-test on mean catch data of the respective phases. We evaluated the strength of the time-lagged relationship between IOD and Chl-a, as well as between Chl-a and small pelagic fish production, implementing the statistical cross-correlation analysis.

## 3. Results

### 3.1. Temperature, Salinity, and Chl-a (In Situ Observations)

Fluctuations in water surface elevation concurrent with variation in the dipole mode index (DMI) during 2019 indicated the beginning of the pIOD in late June and reaching its peak in October and ending in December, and the strikingly different vertical profiles of temperature and salinity within Palabuhanratu Bay corresponding with the peak (October) and the end (December) depict intense upwelling during the peak phase (Figure 3). The mean sea surface temperature (SST) observed at the five sampling stations was distinctly lower (23.35 °C) in the peak of pIOD phase compared to the end of pIOD phase (29.15 °C) with a clear difference of close to 6.00 °C. At a depth of 15 to 25 m, the temperature difference was even higher compared to the SST, which was around 7.00 °C. The thermocline layer rises to the shallower depth layer up to 80 m at the peak of the pIOD phase. The mean of surface salinity observed at all five sampling stations was 34.77 practical salinity units (psu) at the peak of the pIOD phase, while at the end of the pIOD it was 33.41 psu. The difference in surface salinity between the peak and the end of the pIOD phase was distinctly higher (1.36 psu). Furthermore, the vertical distribution of temperature and salinity clearly indicates an intense upwelling during the 2019 pIOD.

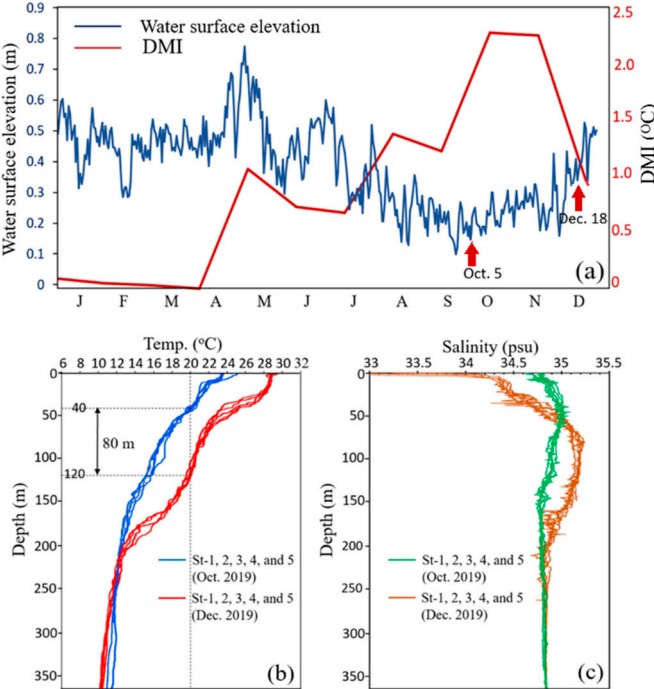

**Figure 3.** (**a**) Water surface elevation concurrent with variation in the dipole mode index 2019, (**b**) temperature, and (**c**) salinity profile observed on October 5 (the peak of positive Indian Ocean Dipole) and on December 18 (the end of positive Indian Ocean Dipole), 2019, within the Palabuhanratu Bay.

The mean Chl-a concentration at the peak of the pIOD phase in the Palabuhanratu Bay was 5.77 mg/m$^3$, increased six-fold compared to the end of the pIOD phase (0.95 mg/m$^3$) (Table 1). The high Chl-a concentration corresponds with a very strong upwelling in the study area during the 2019 pIOD phase.

**Table 1.** The chlorophyll-a concentration measured at five sampling stations in the Palabuhanratu Bay at the peak and end of the 2019 positive Indian Ocean Dipole phase.

| Stations | Chl-a (mg/m$^3$) | |
|---|---|---|
| | Peak of pIOD | End of pIOD |
| St-1 (7.00° S 106.45° E) | 0.42 | 0.62 |
| St-2 (7.03° S 106.45° E) | 5.29 | 0.81 |
| St-3 (7.05° S 106.42° E) | 5.03 | 1.28 |
| St-4 (7.07° S 106.38° E) | 8.06 | 1.15 |
| St-5 (7.12° S 106.31° E) | 10.06 | 0.90 |
| Mean | 5.77 | 0.95 |

*3.2. IOD and Interannual Variability of Pelagic Fishery Environment*

We used a Hovmöller diagram of satellite-derived Chl-a and SST time series data of the sampling tract extending from 106° E to 114° E and centered around 7.54° S along the fishing ground off the southern Java coast (Figure 2) to highlight the variability of SST and Chl-a during the pIOD and nIOD phases for 2003–2019 (Figure 4a–c). The time series of Chl-a concentrations and SST depicts the impact of the 2019 pIOD phase and 2016 nIOD phase, along with other such phases during the previous IOD events. Of all pIOD phases during 2003–2019, the pIOD phase of 2019 was noticed to be one of the most intense and was characterized by a strong Chl-a bloom and cooler surface temperature and the enhanced Chl-a bloom for an extended period. The nIOD phase of 2016 was characterized by the lower Chl-a concentration and higher SST.

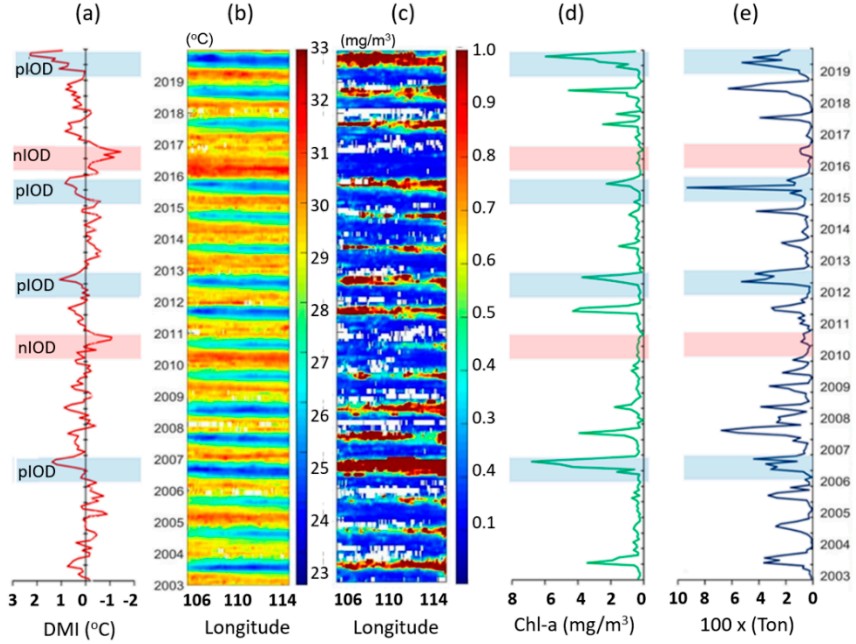

**Figure 4.** (**a**) Monthly time series of the dipole mode index during 2003–2019 showing the positive Indian Ocean Dipole phases (pIOD) and negative Indian Ocean Dipole phases (nIOD) along with the Hovmöller diagram of satellite-derived data of (**b**) sea surface temperature; and (**c**) chlorophyll-a. Monthly time series of (**d**) observed chlorophyll-a in the Eastern Indian Ocean off Java; and (**e**) landing of small pelagic fish catch in the Palabuhanratu fishing port during the same period.

The PSD analysis shows the significant variance in Chl-a over the course of a one-year and a period greater than four years (Figure 5). The dominant peak at one-year represents the monsoon's influence. The peak at more than four-years (51 months) reflects the IOD phenomenon's impact in the EIO.

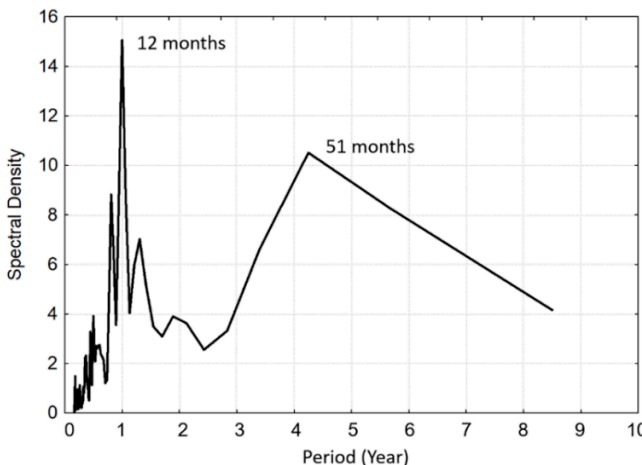

**Figure 5.** Power spectral density of the chlorophyll-a concentration in the Palabuhanratu Bay.

The monthly time series of landings of small pelagic fish are shown in Figure 4e. During the pIOD phase of 2006, 2012, 2015, and 2019, the increase in Chl-a concentration (Figure 4c,d), which is an indicator of phytoplankton biomass, was followed by an increase in small pelagic fish production (Figure 4e). Conversely, during the nIOD phase of 2010 and 2016, the phytoplankton biomass was lower than normal, especially in summer 2010 and 2016, followed by a decrease in the production of small pelagic fish.

### 3.3. Pelagic Fish Catch Variability During 2016 nIOD and 2019 pIOD Phases

Variations in the catches of small pelagic fish in the EIO off Palabuhanratu Bay was observed during the 2016 nIOD and the 2019 pIOD phases. The average of 2014, 2017, and 2018 was considered as the neutral phase. Almost all the kinds of small pelagic caught fish increased during 2019 (Figure 6a). We found that the total catch of small pelagic fish during pIOD phases was twice as large as the neutral condition. Meanwhile, at the time of the 2016 nIOD phase, the catch was five times smaller than that of the neutral condition (Figure 6b).

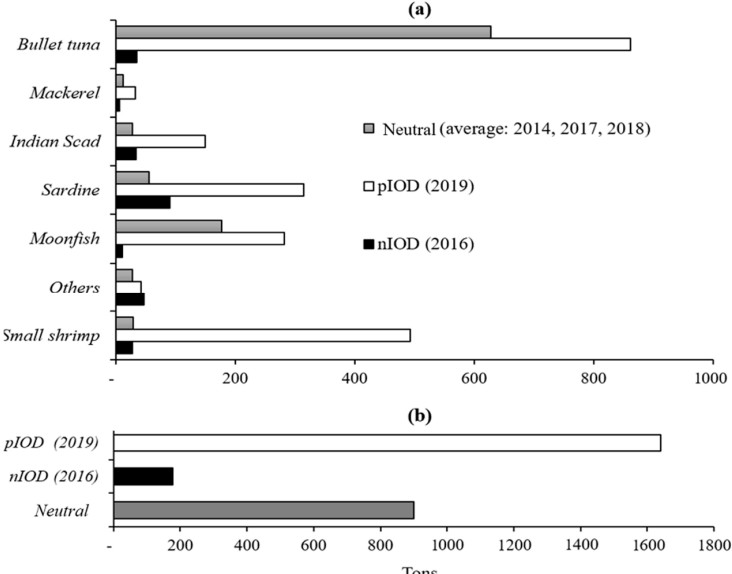

**Figure 6.** Variation in the catches of small pelagic fish (**a**) by species, and (**b**) total catch in the Palabuhanratu Fishing Port during the neutral conditions (average of 2014, 2017, and 2018), Indian Ocean Dipole negative phase (2016) and Indian Ocean Dipole positive phase (2019).

The mean of the quantity of fish catch landing at the Palabuhanratu fishing port during the nIOD phase was 36.08 tons, and during the pIOD phase, it was 310.93 tons. We compared the means of the two samples (catches during the nIOD and pIOD) by using the paired Student's *t*-test. The result indicated a significant difference between the means of catches of small pelagic fish between the 2016 nIOD and 2019 nIOD phases ($p < 0.05$), with considerably higher catches during the 2019 pIOD phase, and, contrastingly, very low catches during the 2016 nIOD phase.

DMI and Chl-a concentration are positively correlated (Figure 7a). The maximum correlation had a time lag of zero. The cross-correlation diagram is asymmetric: correlations with positive time lag are dominant over correlations with a negative time lag. The positive correlation between DMI and Chl-a can be explained by the strong upwelling of nutrient-rich sub-thermocline water into the upper euphotic layer, resulting in an increase in phytoplankton growth rate. Chl-a and small pelagic fish catch correlation are significantly positive from zero up to the two-month time lag, and from 11-months up to the 13-month time lag (Figure 7b).

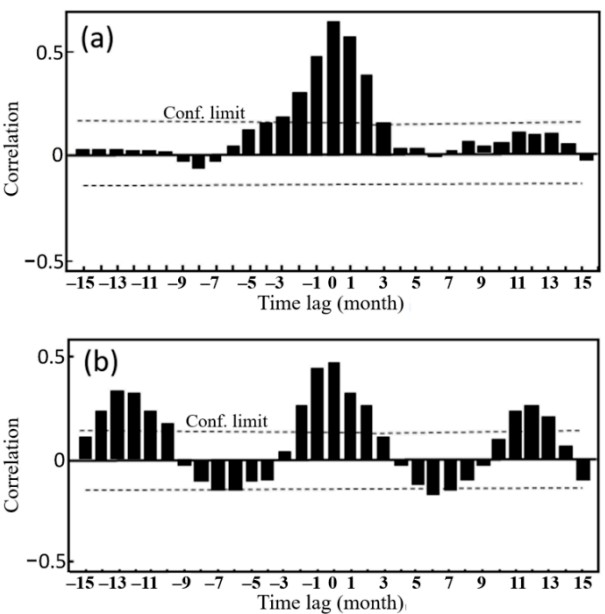

**Figure 7.** Time-lag cross-correlation coefficients between: (**a**) Dipole Mode Index and chlorophyll-a concentration; and (**b**) chlorophyll-a concentration and small pelagic fish landing in the Palabuhanratu Fishing Port.

## 4. Discussion

Researchers have documented coastal upwelling off Java since the earliest oceanographic studies [21]. The coastal upwelling in the EIO off Java has a strong seasonal cycle [27–31]. These previous studies showed that the difference in SST between the southeast monsoon (upwelling) and northwest monsoon (downwelling) was only around 2.00 °C, and thermocline rose up to 40 m in the Palabuhanratu Bay [32]. The fluctuation of water surface elevation, which is an indicator of upwelling (interpreted with concurrent variation in the DMI during January–December of 2019) indicated the beginning of the pIOD in late June, reaching its peak in October, and ending in December (Figure 3a). The differences in the mean SST, thermocline depth, and salinity between the peak and end of the 2019 pIOD phase were 6.00 °C, 80 m, and 1.36 psu, respectively, indicating very strong coastal and open-ocean upwelling during the 2019 pIOD phase (Figure 3b,c).

Our results from the PSD analysis (Figure 5) indicate that the significant variance in the Chl-a in the EIO off Java at the one-year and at more than four years (51 months) period can be attributed to the southwest monsoon's influence and that of IOD phenomenon,

respectively. This finding agrees with previous studies, where the intense Chl-a bloom was positively associated with the pIOD phase [13,15,27].

Based on the Chl-a data derived by satellite sensors (Figure 8) and field observations (Table 1), we described the response of phytoplankton to the strong upwelling events in the EIO off Java from July to October 2019. With the help of in situ Chl-a measurements, we confirmed the significant (five-fold) increase in Chl-a concentration during the peak phase of the 2019 pIOD phase (Table 1) compared to that of the end phase. Previous studies have shown the positive anomalies of Chl-a concentration derived from satellites in EIO off Java during the 1997 and 2006 pIOD phases [14,27]. The upwelling during pIOD phases associated with the surrounding zones is evident from its high Chl-a concentrations [18]. In agreement with the results of Susanto and Mara [20] and Iskandar [13], we found increases in Chl-a concentration in the EIO during the 2019 pIOD phase. In contrast, there was a downwelling in the 2016 nIOD phase that caused a very low Chl-a concentration (Figure 8a).

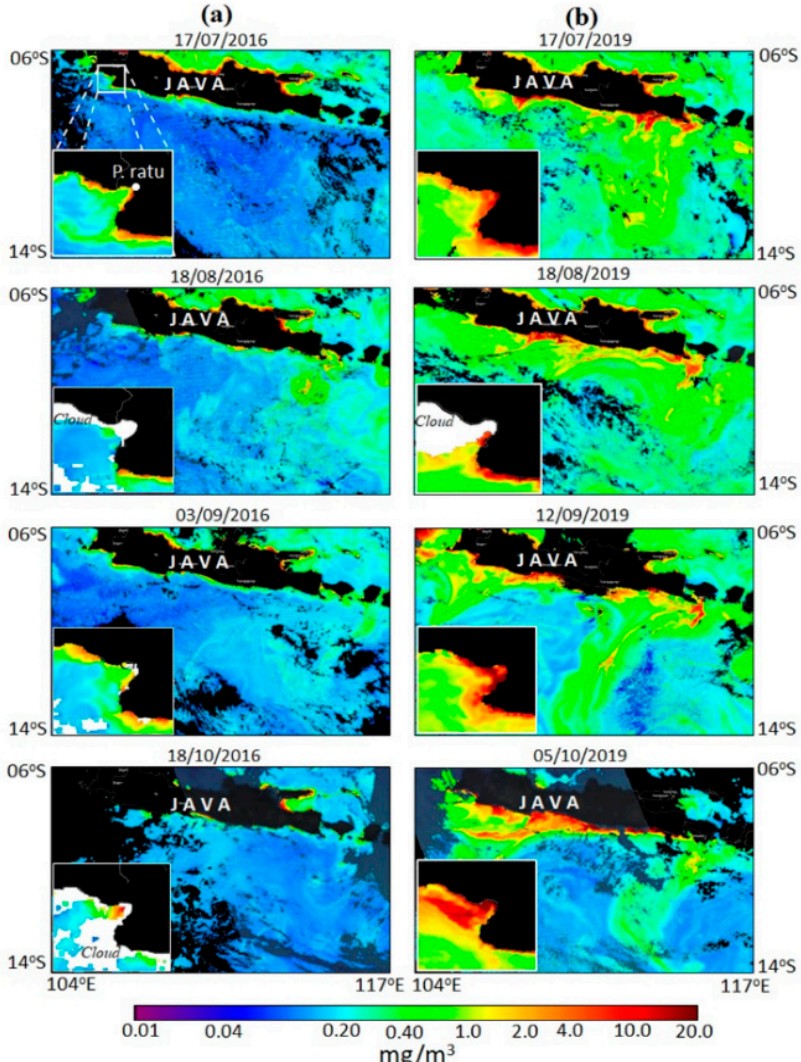

**Figure 8.** Distributions of chlorophyll-a concentration (July–October) in the Eastern Indian Ocean off Java and the Palabuhanratu Bay during: (**a**) the 2016 Indian Ocean Dipole negative phase; and (**b**) the 2019 Indian Ocean Dipole positive phase. The chlorophyll images of each month (July–October) are shown using the daily image of the cloud-free days available during the respective months.

The high Chl-a during the pIOD phases can be explained by an intense upwelling that typically brings cooler water from the deeper depth, rich in nutrients. These nutrients "fertilize" surface waters and trigger high biological productivity in the EIO [13,27].

The 2019 pIOD phase modulated the upwelling, thereby influencing biological productivity and fish catches across the EIO off Java. The total catch of small pelagic fish that landed at the Palabuhanratu Fishery Port during the 2019 pIOD phase was five times greater than that of the 2016 nIOD phase. This can be explained by the food web, because plankton is the main food of small pelagic fish. Particularly in the upwelling regions, there are often a few small plankton-feeding pelagic fish species that dominate in the higher trophic level. Based on the field observations between October and December 2019, the number of small pelagic fish landings increased sharply. We also found many juvenile fish contributing to the fish catch. This is due to coastal transport of the upwelled water and nutrients during the 2019 pIOD phase, injecting plankton-rich waters into key fish spawning areas located south of Java [33]. The pIOD altered the trophodynamics related to the small fish in the pelagic zone in the EIO significantly, due to an increase in the phytoplankton abundance and biomass.

The maximum correlation between Chl-a and small pelagic fish catch had a time lag of zero and was significantly positive up to the two-month period (Figure 7b). Such correlation indicates that phytoplankton abundance directly affects the fish abundance, because they include several kinds of small pelagic fish such as sardinella which are phytoplankton-feeders [34]. Furthermore, the significant cross-correlation between the Chl-a and small pelagic fish catch had a time lag of one-year (11–13 months), which can be attributed to the larvae of fish often reaching their first feeding stage at approximately the same time that phytoplankton blooms occur [35,36]. Strong downwelling (upwelling) during nIOD (pOD) phases causes low (high) Chl-a, which may result in decrease (increase) fish recruitment through the match (mismatch) [37,38], because the key element in larval survival is often the availability of food [39,40]. Availability of plankton in sufficient quantities at the right spatiotemporal scale enhances the larval survival rate. Further investigation is necessary to understand how the enhanced phytoplankton production during pIOD and other relevant environmental drivers interacts with the life cycle and influences the recruitment of the small pelagic fish and the interannual variability of the fish catch, in the Palabuhanratu Bay [37,38].

Along the coast of the EIO off south Java, the small pelagic fish play an important role in the fishers' economics. The fluctuation in prices of small pelagic fish is considerable, owing to the uncertainty of production. When fish were more plentiful during the 2019 pIOD phase but market demand was stable, prices dropped. However, during the 2016 nIOD phase, when the fish catches declined, prices rose. Besides the uncertainty of fish production, the perishable nature of fish also affects the price thereof [41,42].

The IOD-related fish price variability can be maintained by regulating the fishing fleets and fish catch distribution. We know well that during the pIOD (nIOD) phases, the production of small pelagic fish was very high (low) because of the variability of Chl-a concentrations; as a result, there were increases (decreases) in the primary production in the waters. As the IOD phases can be predicted a few months prior, so that fish catches which are plentiful during the pIOD phase can be distributed to other regions to prevent disequilibrium in market demand and supply. Likewise, during the nIOD phase, fish supplies from other regions can be efficiently transported. During pIOD phases, fishing operations can also be reduced, and juvenile-sized fishing prohibited. If the market demand and supply are in equilibrium, the fish price will be stable because the fish supply is highly elastic [42].

## 5. Conclusions

During the interannual cycle of the 2019 pIOD phase, there was particularly strong coastal upwelling, as seen from the six-fold difference in both temperature and Chl-a concentration between the peak and end of the pIOD in the Palabuhanratu Bay. Conversely,

during the 2016 nIOD phase, there was a strong downwelling. During pIOD (nIOD) phases, the Chl-a concentration became abnormally high (low) over the EIO coastal waters off Java, through the increased (decreased) intensity of upwelling (downwelling). Chl-a bloom (decline) during the strong coastal upwelling (downwelling) subsequently increases (decreases) the abundance of small pelagic fish in the EIO off Java's coastal waters.

Variation in Chl-a concentrations due to IOD events significantly affects the small pelagic fish resources in the EIO off Java's coastal waters. Therefore, the information or predictions of both pIOD and nIOD phases can be used as a basis for fishing management to increase fishermen's income and reduce risks associated with the fisheries sector during the IOD phases.

**Author Contributions:** Conceptualization, J.L.-G., K.M., I.W.N., L.A. and E.S.; methodology, J.L.-G., K.M. and I.W.N.; validation, J.L.-G., M.T.H. and E.M.; formal analysis, J.L.-G., and T.O.; investigation, J.L.-G., N.M.N.N. and E.M.; data curation, J.L.-G., N.M.N.N., M.T.H., B.M.K.R. and A.P.; writing—original draft preparation, J.L.-G.; writing—review and editing, J.L.-G., E.S., K.M., N.M.N.N. and I.W.N.; visualization, J.L.-G. and H.A.R.; project administration, J.L-G.; funding acquisition, E.S. All authors have read and agreed to the published version of the manuscript.

**Funding:** This research was funded by the Ministry of Research Technology and Higher Education of Indonesia through the "Competitive Magister Research", grant number (1/E1/KP.PTNBH/2020 and 1/AMD/E1/KP.PTNBH/2020). The Open Access and Article Processing Charge (APC) was funded by the Asia-Pacific Network for Global Change Research (CAF2017-RR02-CMY-Siswanto).

**Acknowledgments:** The authors thank the anonymous reviewers for their valuable and constructive comments. Thanks to Saji Hameed for encouraging us to conduct a marine survey in the positive phase of IOD 2019. I thank the NASA Ocean Color Project, NASA Earth Data Open Access for Sciences and Asia-Pacific Data Research-Center (APDRC) for processing and distributing the datasets. Thanks are also due to "PPS Palabuhanratu" for providing fishery statistical data (2003–2019).

**Conflicts of Interest:** The authors declare no conflict of interest. The funders had no role in the design of the study; in the collection, analyses, or interpretation of data; in the writing of the manuscript, or in the decision to publish the results.

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
