# Peer review of "Impact of the Strong Downwelling (Upwelling) on Small Pelagic Fish Production during the 2016 (2019) Negative (Positive) Indian Ocean Dipole Events in the Eastern Indian Ocean off Java"

_climate, doi:10.3390/cli9020029_

Round 1

Reviewer 1 Report

Accept with minor corrections

Revision of Manuscript entitled "Influence of the Strong Downwelling (Upwelling) Impact on Small Pelagic Fish Production During the  2016 (2019) Negative (Positive) Indian Ocean Dipole  Events in the Eastern Indian Ocean off Java"

General comments

This manuscript analyzed IOD variability on Chl-a conc. (as an indicator of phytoplankton biomass) and small pelagic fish production in the EIO off Java. Field campaign for data collection was complemented with the use of both ground-based and satellite-based datasets to provide deep insights into how IOD variability at inter-annual time scales influences biological productivity and fish productions across the Indian Ocean.  

In summary, the negative (positive) IOD episode of 2016 (2019) was a primary climate driver for large portions of Chl-a conc. The data analysis showed that it is very likely that IOD-related forcing (i.e. downwelling (upwelling)) has affected preponderant decrease (increase) of small pelagic fish production in the region. From a forecast viewpoint, the study analyses has created a confidence our ability to forecast and predict IOD phenomena and its associated impacts, as the result findings have suggested. The manuscript has provide several data sources and statistical tools useful for relevant stakeholders to replicate for purposes of monitoring and evaluation.

The manuscript report of the data analysis was not complicated and presents interesting and well written results. The statistical analysis used (correlation and student-t test) are standard and widely used in the field of climate science. The general public may find this work interesting, and particularly contribute to the on-debate on climate change and its impact on ocean ecology. The paper has a good flow, holds the reader's attention. However I observed few inconsistencies in reporting figures and formatting.

Overall, the manuscript provide significant contribution to the field of climate research (specifically ocean-atmosphere interactions) and merit acceptance for publication with minor revision (see detailed comments below).

a)      Title:

Overall, the title is appropriate for the paper

b)     Abstract:

Overall, The abstract is well written and informative.

Minor comment

Although researchers have investigated the impact of Indian Ocean Dipole (IOD) phases on human lives, only a few have examined such impacts on fisheries.

Line 36: decreases

c)      Keywords:

Six (6) keywords were provided by the authors. It is within the journal’s specified number and the keyword are ok except that three keywords appeared in the title (i.e. upwelling downwelling, small pelagic fish).

I would recommend that to help the article search results in the future or increase the articles visibility to a large audience, keywords should be different from the title; the authors replace these keywords or include two additional keywords.

d)     Introduction:

The introduction is well written. The references used in this manuscript seems to be very old. In many cases, references as old as 1990s and early 2000s were cited.  However, I think this section would be significantly improved with the addition of more current references.

Minor comments

Line 43: IOD positive phases

Line 45: delete “clearly”

Line 47-48: Researchers have widely investigated the impact of IOD phases on human lives . They have …

Line 63: Delete “and” and start the sentence as “Conversely, it tend …

Line 70: delete “so as”

Line 73: “dan” should be “and”

e)      Abbreviations:

All abbreviations are defined at first use except those appearing in Table and Figure captions

f)       Methodology:

Overall, the methodology used in this section is acceptable. The research procedures and techniques used are standard for this scientific research. The references of all data used were duly acknowledged, and analyses are clearly defined, straight forward, and is reproducible for in-situ data used. For example, field visits, specifics dates, instruments deployment and use were explained.

The statistical analysis used (monthly variability, correlation, were explained within the context of the study objectives and hypothesis. Based on the hypothesis, I think the student-t test were testable and helped to explain the objectives. However, the description for satellite based Chl-a and SST was too general and vague. Important information on spatial resolution was missing here. No information on how these datasets were processed. The author’s scope and capability is limited by resources availability etc. however, no information was clearly provided for assumption(s) and the limitation(s) in this study.

The following minor comments need to be addressed here.

References of data sources not consistent with mpdi style of referencing. See line 90 and 92.

Data analysis section: The explanation of the data collection process is detailed excerpt for some missing information on satellite-based Chl-a and SST. For example, specify the type and specification of satellite-based Chl-a and SST used. The temporal and spatial of the satellite data should be specified.

Line 88-90: missing information.

Line 88-90: “The monthly and daily means of satellite derived Chl-a concentration and SST used in this study are from https://oceancolor.gsfc.nasa.gov/ and https://worldview.earthdata.nasa.gov/, respectively.” Authors need to include more information the period of data used. For example was the temporal resolution (or period used) same for both Chl-a conc. and SST? Additional information on the spatial resolution of the satellite based Chl-a conc. and SST is required here. For example, multiple sensors maybe used to acquire Chl-a. It is unclear what spatial resolution and type of sensor for Chl-a were used.

Line 89: insert hyphen “satellite-derived”.

Line 98: Delete “There” and start the sentence “We collected ….”

Line 101: insert article “… strength of the time-lagged  ...”

Line 102: insert “and” “… production, and we performed …”

Line 100: word missing “paired student t-test”.

Figure 2. Missing scale bar and no footnotes for  St- to St-5 description

g)     Results

The results are clear, well presented, and provide a basis for developing recommendations to assist policymakers and other stakeholders. This section is overall well complemented with clear tables and figures that help to visualize the results. However, the format results need to be re-organized especially subsection’s captions in results section were poorly done.

Minor corrections

Line 109, 110, 112, 121, 188: make for the degrees symbol superscript oC

Line 112: maintain consistency for decimals places. For example, on line 111, you have two decimal places for 6.00 oC but in the next line 7 oC has no decimal places.

Line 112-3: Results: Lack of consistency in use of meters and “m”. “m” is used here, however, in the section 2: Materials and Methods, you used “meters” in full.

Line 114: psu???

Line 121: m3 should be m3. The cube should be superscript.

Table 1 and Figure 1: No footnote is provided for the stations (St-1 etc.) numbering

Line 121: Double labelling of 3.1. See line 106.

Figure 4: Inconsistency in caption labelling. Hovmoller diagram for Figure 4b and c captions are not well labelled and positioned.

Figure 4: Define all abbreviations/acroymns here. pIOD, nIOP, EIO, SST, and Chl-a here also.

Line 153: numbering of caption of subsection 3.1 wrongly labeled again.

Line 155: insert space for 2014, 2017. Reduce space in “… phase, in”.

Line 161: One decimal place??? In previous sections above you maintain two decimal places.

Figure 7: include and write abbreviations in full here also for DMI and Chl-a.

Line 160-166: Results of student-t students were reported. Where is the table showing the detail results. I think there are missing

Line 110: delete “so”

Line 112: replace “compared to” with “than”

Line 113: replace “the peak of the pIOD phase” with pIOD phase’s peak. The mean of surface salinity at pIOD phase’s peak is …”

Line 126: delete “as a way of plotting” to read “We used a Hovmoller to plot”

Line 127: add “s” to read “highlights”

Line 131: replace “over the course of” with “for”

Line 132: replace “is a reference to” with “references”

Line 133: monsoons ; insert “comma” and replace “which reflects” @ “…more than 4 years, reflecting the …”

 Line 133-134: replace in accordance with a number of” with “following several” to read “This finding is following several previous …

Line 138: satellite-derived

Line 145: insert full stop after EIO [10,20].

Line 145-146: Should read “The IOD has altered the small pelagic zone's trophodynamics in the EIO due to an increase in the phytoplankton abundance and biomass significantly.”

Line 149: delete “was” and “by”

Line 151: delete “which was” and replace “a” with “the”

Line 154-155: Should read “Variations in …. pIOD phases. The average …”

Line 158: replace “compared to” with “than”

Line 161: “ By using a student t test”

Line 161-163: Consider revising the sentence.

Line 170-171: Should read “The maximum correlation has a time lag of zero. The cross-correlation diagram is asymmetric: Correlations with a positive time lag dominate over correlations with a negative time lag.”

Line 175-176: Should read “Chl-a and small pelagic fishes' correlation is significantly positive from zero up to the 2-month time lag (Figure 7b).”

h)     Discussion

In Line 183-189, the authors provided insightful content and context of previous studies on upwelling and downwelling in the region which provided the basis for discussing the result findings of Chl-a and phytoplankton relationship. Although, it is reasonable to do that, I am wondering why the authors did not made use of available in-situ or satellite based data on sea surface height to assess the magnitude of the upwelling and downwelling phenomenon in the region?

Minor comments

Line 186: insert comma after “2oC, and delete “up”. Insert “the” to read “in the Palabuhanratu Bay

Line 188: Inconsistency; “6” without decimals. Use “m” should be consistent with use elsewhere in the manuscript.

Figure 8: include and write abbreviation in full

Line 190: Should read “ Based on …”

Line 197-198: Should read “In contrast, there was a downwelling in the 2016 nIOD phase that caused a very low Chl-a concentration (Figure 8a).”

Line 203: Should read “The 2019 pIOD phase's strength modulates the upwelling, thereby influencing biological productivity and fish catches across the EIO off Java.”

Line 210-212: Should read “The IOD has altered the small pelagic zone's trophodynamics in the EIO due to an increase in the phytoplankton abundance and biomass significantly.”

Line 221: change “was” to “were”

Conclusion

This section is well written. The conclusions were based on the result findings and logically stated. However, I suggest the authors can link the results findings to policy implication and provide recommendation. Also, the recommendation for a follow-up study maybe a strong point in this paper.

Minor comments

Line 234: upwelling’s

References

Many references in the text are too old. The authors need to infuse current references.

Recommendation

Accept with minor revision

Author Response

 First of all, we wish to thank the reviewers for going through the manuscript carefully, appreciating the actual content of the manuscript, and providing detailed
comments/suggestions for further improvement in the content. We have tried to provide the item reply to all the points raised by the reviewers (attached file).

Sincerely yours,

Jonson

Reviewer 2 Report

The study from Lumban-Gaol et al., deals with fish landings variability and environmental condition in the Eastern Indian Ocean off Java. The study is interesting, the statistical design is great and the results support the discussions.

My main concern is about the time-lag cross correlation. I think that this is important to talk about the significant correlation about 12 months. To me, this may stress that years with high (low) chl a may increase (decrease) fish recruitment through match mismatch and explain why landings 1 year later are more important (this is particularly true for small pelagic fish with rapid growth). I think that 2 or 3 sentences in the discussion about this aspect and discussing what mechanisms can improve recruitment and future landings (using the most recent reference about pelagic fish recruitment, i.e., Ferreira et al., 2020 and Brosset et al., 2020) will strengthen the discussion.

My other comments are minor:

Please, be careful with the °C sign which is often noted oC in the manuscript.

The lines 161-163 about statistic are not needed here. Please just say that the p value is under 0.05 and that the difference is significant.

The reference to be included (both are in open access):

Ferreira, A. S. A., Stige, L. C., Neuheimer, A. B., Bogstad, B., Yaragina, N., Prokopchuk, I., & Durant, J. M. (2020). Match− mismatch dynamics in the Norwegian− Barents Sea system. Marine Ecology Progress Series, LFCav5.

P Brosset, AD Smith, S Plourde, M Castonguay, C Lehoux, ... A fine-scale multi-step approach to understand fish recruitment variability. Scientific Reports 10 (1), 1-14

Author Response

Dear Reviewer,

 First of all, we wish to thank the reviewers for going through the manuscript carefully, appreciating the actual content of the manuscript, and providing detailed
comments/suggestions for further improvement in the content. We have tried to provide the item reply to all the points raised by the reviewers (attached file).

Sincerely yours,

Jonson

This manuscript is a resubmission of an earlier submission. The following is a list of the peer review reports and author responses from that submission.